# *Ferula communis* L. (*Apiaceae*) Root Acetone-Water Extract: Phytochemical Analysis, Cytotoxicity and In Vitro Evaluation of Estrogenic Properties

**DOI:** 10.3390/plants11151905

**Published:** 2022-07-22

**Authors:** Jessica Maiuolo, Vincenzo Musolino, Lorenza Guarnieri, Roberta Macrì, Anna Rita Coppoletta, Antonio Cardamone, Maria Serra, Micaela Gliozzi, Irene Bava, Carmine Lupia, Luigi Tucci, Ezio Bombardelli, Vincenzo Mollace

**Affiliations:** 1Laboratory of Pharmaceutical Biology, IRC-FSH Center, Department of Health Sciences, University “Magna Græcia” of Catanzaro, 88100 Catanzaro, Italy; v.musolino@unicz.it (V.M.); lupia10@libero.it (C.L.); 2IRC-FSH Center, Department of Health Sciences, University “Magna Græcia” of Catanzaro, 88100 Catanzaro, Italy; lorenzacz808@gmail.com (L.G.); robertamacri85@gmail.com (R.M.); annarita.coppoletta@libero.it (A.R.C.); tony.c@outlook.it (A.C.); maria.serra@studenti.unicz.it (M.S.); gliozzi@unicz.it (M.G.); irenebava@libero.it (I.B.); l.tucci@head-sa.com (L.T.); 3Plantex E C. Sas. Galleria Unione, 5, 20122 Milano, Italy; ezio.bombardelli@plantexresearch.it; 4Nutramed S.c.a.r.l., Complesso Ninì Barbieri, Roccelletta di Borgia, 88021 Catanzaro, Italy; 5IRCCS San Raffaele, Via di Valcannuta 247, 00133 Rome, Italy

**Keywords:** *Ferula communis* L. extract, ferutinin, ferulenol, MCF-7 cells, HeLa cells, Saos-2 cells

## Abstract

*Ferula communis* L. (*F. communis*) belongs to the Apiaceae family and is a herbaceous plant with various pharmaceutical properties, due to the different contents of bioactive compounds extracted mainly from its roots, as well as its leaves and rhizome. To date, this plant extract has demonstrated estrogenic, anti-inflammatory, antiproliferative, cytotoxic, antimicrobial and anti-neoplastic properties. Its estrogenic activity is justified by the presence of ferutinin, an ester of a sesquiterpenic alcohol that acts as an agonist for estrogen receptors, with a chemical formula equal to C_22_H_3_O_4_. The component present in *F. communis* responsible for the toxicity of the plant is ferulenol, a prenylated coumarin with the chemical formula C_24_H_30_O_3_. This compound is capable of inducing mortality via its strong anti-coagulant properties, leading to a lethal hemorrhagic syndrome, ferulosis, in animals that feed on a chemotype of *F. communis* containing a high amount of ferulenol. The removal of the component ferulenol makes extracts of *Ferula* non-toxic. In fact, the remaining prenylated coumarins are not present in concentrations sufficient to induce toxicity. The intake of high concentrations of the extract of this plant leads a double dose-dependent effect that is typical of sesquiterpenes such as ferutinin. Here, we assessed the cytotoxicity and the estrogenic properties of the *F. communis* phytocomplex obtained through extraction using a mixture of acetone and water. Among the active constituents of *F. communis*, the identification of ferutinin and ferulenol was performed using HPLC. The effects of the extract were evaluated, following the removal of ferulenol, on three cell lines: human breast cancer MCF-7, human cervical cancer HeLa and human osteoblastic sarcoma Saos-2. The choice of these cell lines was justified by the need to mimic certain processes which may occur in vivo and which are estrogen-dependent. The obtained results demonstrated that *F. communis* extract, in addition to possessing an estrogenic-like property, showed a dose-dependent effect. Low concentrations (0.1–0.8 μM) demonstrated a hyperproliferative effect, whereas higher concentrations (1.6–50 μM) were toxic. Therefore, this extract could be an excellent candidate to make up for a reduction or lack of estrogen.

## 1. Introduction

In recent decades, the study of plant extracts has increased exponentially due to their countless beneficial effects on human health. In fact, accumulated evidence, conducted both in vitro and in vivo, has pointed out that most compounds of plant origin possess antioxidant, anti-inflammatory, and anticancer properties, among others [1,2,3]. The genus *Ferula* L. belongs to the family of the *Apiaceae* and is represented by about 170 species mostly present in all of Central Asia, the Mediterranean region and North Africa [4]. From the ancient Latin name, used to indicate a straight stem plant, three species have been described in Italy: *Ferula communis* L., *Ferula glauca* L. and *Ferula arrigonii* B [5,6,7]. In this genus, several compounds have been identified with various effects such as sesquiterpenes [8], sesquiterpene coumarins [9], sesquiterpene lactones [10] and sulfur-containing compounds [11]. This herbaceous plant has different pharmaceutical properties due to the different content of bioactive compounds extracted mainly from the roots, but also from the leaves and rhizome. *F. communis*, traditionally used to treat fever, dysentery and skin infections, has shown estrogenic, anti-inflammatory, antiproliferative, cytotoxic, antimicrobial, and anti-neoplastic activities [12,13,14]. In particular, estrogenic property is justified by the presence of ferutinin, an ester of a sesquiterpenic alcohol that acts as an agonist for estrogen receptor, of a chemical formula equal to C_22_H_30_O_4_: ferutinin is responsible for most of the effects produced by *F. communis* extract administration. On the contrary, a poisonous chemotype of *F. communis* exists, represented in a high percentage of prenylated coumarins that perform toxicity. Among these, ferulenol, whose chemical formula is C_24_H_30_O_3_, is the most represented and, with its strong anti-coagulant properties, leads to a lethal hemorrhagic syndrome in animals (goats, sheep, cattle and horses). This disease is known as ferulosis and affects animals that feed on this toxic chemotype of *F. communis* [15]. The chemical structures of ferutinin and ferulenol are represented in Figure 1a. Regardless of the extract of *F. communis*, it will be possible to reduce its toxicity by eliminating the content of prenylated coumarin ferulenol. Several pieces of data highlighted that *F. communis* extract (FER-E), completely deprived of ferulenol, could be an effective alternative hormone therapy to prevent and treat post-menopausal symptoms thanks to it having a high amount of the sesquiterpene ferutinin [16]. In fact, this compound is able to mime the endocrine activity of the ovaries [17]. In addition, the ferutinin structure is similar to that of steroid hormones and has been classified as a phytoestrogen with an affinity for both subtypes of estrogen receptors [18,19]. This article is divided into two parts: first, the obtaining of the extract FER-E and its characterization are described. Subsequently, the effects of FER-E will be evaluated on three cell lines that mimic certain effects found, in vivo, during menopause: the MCF-7 human breast cancer line, the HeLa human cervical cancer line and the Saos-2 human osteoblastic sarcoma line. Before discussing the effects of these cell lines, it would be advisable to know what is published in the literature about it: to date, exposure to estrogen is considered a risk factor for the onset of breast cancer because it is able to promote breast cell proliferation. In particular, the effects of 17-β-estradiol (17-β-E2) occur as a result of binding to its nuclear receptors (RE α and RE β), conformational change and subsequent interaction with chromatin and gene regulation [20,21]. In fact, the inhibition of the estrogenic receptors by treatment with tamoxifen, a substance that inhibits the binding of estrogen to its receptor, or ICI 182.780, an estrogen receptor antagonist, is considered one of the main strategies for the prevention and treatment of breast cancer [22]. Studies conducted in vitro with the use of ferutinin, a compound extracted from Ferula showed that it has the property of inhibiting and suppressing the growth of different tumor cell lines [12,23,24,25]. In addition, in vivo data have shown that it does not promote the development of systemic toxicity [26]. Ferutinin is considered a selective estrogen receptor modulator, acting as an ER α agonist (IC50 = 33.1 nM) and an agonist/antagonist to ERβ (IC50 = 180.5 nM) [27,28]. Safi et al. have demonstrated that ferutin and its hemi-Synthetic analogue jaesckeanadiol-3-p-hydroxyphenylpropanoate induce a marked cytotoxic effect towards the MCF7 tumor line, reducing cell proliferation, altering the cell cycle with a blockage in the G0/G1 phase and inducing apoptotic death [29]. In addition, ferutinin has been shown to damage DNA, not only in MCF7 cells, but also in normal fibroblasts (HFF3); its effect (evaluated with IC50) was more effective than the well-known anticancer drugs doxorubicin and vincristine, which often undergo drug-resistance reactions. Therefore, these findings suggest that ferutinin could be considered an effective anticancer agent for future in vivo and clinical experiments [30]. As FER-E contains ferutinin, can the whole extract enjoy these properties?

On the other hand, if estrogens are a risk factor for breast cancer, it is also true that suspension of estrogen production during physiological menopause (interruption of the menstrual cycle and reduction/suspension of the secretion of estrogen and progesterone ovarian hormones) and induced menopause (menopause caused by a medical pharmacological treatment), also carries the risk of developing breast and ovarian cancer [31]. In fact, Heer et al. showed that post-menopausal women had a statistically much higher incidence of breast cancer than pre-menopausal women in 44 countries worldwide [32]. The increased incidence of breast cancer in post-menopausal women is also justified by the presence of certain risk factors, such as a high abdominal adiposity and increasing body-mass index, both due to lifestyle (physical inactivity, alcohol consumption) [33]. It has also been highlighted that an early onset of menopause is related to an increased risk of Cardio Vascular Diseases (CVD), osteoporosis, depression, a higher prevalence of hormone-related cancers (endometrial, ovarian and breast) and mortality [34]. A cytotoxic effect has been demonstrated by ferutinin and other terpenoids (extracted from the roots of three species of Ferula: *Ferula latisecta*, *Ferula ovina* and *Ferula flabelliloba*) also on the human cancer cell line HeLa [35]. Finally, it is well known that women, during the transition and after the menopause, develop radical changes in bone health, due to reduced bone mineral density, and a high risk of onset of osteoporosis [36]. In fact, bone alterations in menopause, occur not only at the macrostructural level but also at microstructural level, with thinned trabeculae and diminished connectivity. In particular, the decrease in estrogen leads to increased bone turnover with progressive loss of bone mass due to a bone resorption process that is higher than that of the new formation. Bone mass appears to be thinning as estradiol levels decrease and there is a temporal correlation between the onset of vasomotor symptoms (hot flashes and night sweats) and changes in bone health [37]. In particular, ferutin has been shown to induce proliferation and differentiation in human osteoblastic cells [38,39,40]. Hormone replacement therapy (HRT) is prescribed for the purpose of reducing menopause symptoms [41] and to date, it is known that a well-balanced hormone therapy (estrogen–progesterone) is considered potentially beneficial if started within 10 years from the beginning of menopause [42,43]. HRT treatment for more than 5 years has been shown to significantly increase the risk of developing breast cancer, endometrial cancer, ovarian cancer, blood clots, strokes, gallbladder diseases [44,45,46]. Phytoestrogens are plant compounds with estrogen-like properties and they exert an estrogenic or antiestrogenic effect depending on the circulating estrogen level [47]. The use of phytoestrogens to treat the symptoms of menopause could be convincing thanks to the reduced side effects that can develop even after a prolonged treatment. In light of what has been shown, the three named cell lines will be treated with FER-E in order to gain understanding into how the use of *F. communis* can improve human health in pathological conditions of cancer and some districts involved during menopause.

## 2. Results

### 2.1. HPLC Assessment of the Chemical Composition of FER-E

A HPLC analysis was performed following extraction of FER-E to assess the presence of ferutinin and the toxic ferulenol component. As can be observed from the spectrum (shown in Figure 1b) and the quantitative report obtained (Figure 1c), ferutinin is present in high quantities in the extract and constitutes the most represented component. Nevertheless, ferulenol is also present, as indicated by the red arrow, which is responsible for the toxicity of the extract.

Subsequently, ferulenol was removed from the extract, as demonstrated by the HPLC spectrum shown in Figure 2, where the peak for this component is absent. The ferulenol removal method was kindly carried out by Plantex E C. Sas (Galleria unione, 5, 20122, Milan, Italy).

### 2.2. Treatment with 17-β-Estradiol Increases Cell Proliferation in MCF-7, HeLa and Saos-2 Lines

First, to assess whether FER-E has estrogenic effects, we treated cell lines (MCF-7, HeLa and Saos-2) with 17-β-E2 several times (24, 48 and 72 h). In particular, the pattern of increased cell proliferation was similar for all three cell lines (Figure 3a–c). In MCF-7 and HeLa cells, we can appreciate a significant increase in cell proliferation already after 24 h of treatment with 17-β-E2 to concentrations of 1 and 10 μM (*p* < 0.05). After 48 h of treatment, MCF-7 cells demonstrated more significant values with 17-β-E2 to concentrations of 0.01, 0.1 μM (*p* < 0.01) and with 17-β-E2 to concentrations of 1 and 10 μM (*p* < 0.001). For HeLa cells, the treatment with 17-β-E2 0.01–10 μM demonstrated significant results (*p* < 0.001). Finally, treatment with 17-β-E2 (0.01–10 μM) per 72 h generated, both in MCF-7 and HeLa, a proliferative highly significant difference compared to the result in untreated cells (*p* < 0.001). The Saos-2 cell line, although showing a proliferative increase following treatment with 17-β-E2, highlighted lower values: at 24 h, only the concentration of 10 μM was significant (*p* < 0.05); at 48 h, it was the concentrations of 1 and 10 μM (*p* < 0.05 and *p* < 0.01, respectively); at 72 h, it was the concentrations of 0.1, 1 and 10 μM (*p* < 0.05; *p* < 0.01 and *p* < 0.001, respectively). Concentrations above 10 μM (50 and 100 μM) were tested, but the hyperproliferative effect was reduced to 50 μM until completely nullified at the 100 μM concentration. This trend was found to be present in all the cell lines considered after 24 h of treatment. The prolonged treatment for a longer time resulted in a hyperproliferative cancellation after 48 h and this was particularly noticeable after 72 h (data not shown).

### 2.3. FER-E Has a Dose-Dependent Effect: Low Concentrations Stimulate Hyperproliferation, While Higher Concentrations Are Toxic

Subsequently, the three cell lines were treated with increasing concentrations of FER-E for 24, 48 and 72 h. As can be seen in Figure 4a–c, FER-E showed an hyperproliferation effect, mimicking the effect obtained with 17-β-E2. This extract showed a dose-dependent effect: low concentrations (0.1–0.8 μM) showed a hyperproliferative effect, while higher concentrations (1.6–50 μM) were toxic. This double effect is typical of sesquiterpenes, as with ferutinin, and seems to be justified by the modulation, which is sesquitepene-induced, of the permeability of the biological membranes to cationic species [48]. The result was similar in the cell lines considered (MCF-7, HeLa and Saos-2) and for all treatment times (24, 48 and 72 h). In particular, the 1.6 μM concentration has always been shown to be the dividing line between the hyperproliferative and the toxic effect.

In order to confirm the toxic effect, specific mortality experiments were carried out in which cells were exposed to all FER-E concentrations considered and, again, cell mortality was appreciated at concentrations equal to and greater than 1.6 μM, as indicated in Figure 5a–c. The increase in cell mortality, which is concentration-dependent, was statistically significant.

### 2.4. Assessment of the Type of Cell Death, Induced by FER-E, through the V-PI Annexin Test

To determine whether FER-E treatment resulted in apoptotic or necrotic death, the three cell lines were stained with Annexin V-FITC and PI and the study was conducted using flow cytometry. In Figure 6, the results for the cell lines MCF-7, HeLa and Saos-2 are shown in panels a, b, and c, respectively. Low concentrations of FER-E (0.2–0.8 μM) showed no cell distress and cells were annexinV-negative/PI-negative, as were untreated cells. Cells treated with FER-E, starting from the concentration 1.6 μM, were found to be annexin V-positive/Pi-positive, indicating a post-apoptotic death that has become greater, increasing the concentration of FER-E. A strong cellular necrosis (V-negative/PI-positive cells) appeared at the concentration of FER-E 25 μM, showing substantial damage to cells. In the Saos-2 cell line, it was not possible to measure with FER-E 25 μM treatment, since the cells, at this concentration, were all dead. Finally, Figure 7 shows the percentage of florescent cells after treatment with FER-E at all concentrations considered.

## 3. Discussion

At the moment, it is known that the biological effects of *F. communis* on human health depend on the chemotype used: the poisonous chemotype, containing mainly prenyl coumarins such as ferulenol, is able to generate intoxications, ferulosis and also cause death [49,50], while the non-poisonous chemotype, which contains daucane esters whose main component is ferutinin, is traditionally used for its hormonal effects and has been classified as a phytoestrogen [51,52].

In addition to coumarin sesquiterpene [53], sesquiterpenes [54], sesquiterpene coumarin glycosides [55] and sulfur-containing compounds [56], Ferula also contains promising bioactive compounds such as auraptene [57] (with antihypertensive, anti-inflammatory and oncological chemopreventive properties), umbelliprenin [58], (with anti-inflammatory and oncological chemopreventive activities) and galbanic acid [59] (with anticancer and antiangiogenic effects). The extract FER-E, used in this experimental work, was obtained from the root of *F. communis* and was deprived of the toxic component ferulenol; its main constituents are daucane sesquiterpene esters including ferutinin, lapiferin and teferin. In particular, a chromatographic analysis of FER-E has allowed the estimation of titre pairs to 25% in ferutinin. The effects of FER-E were evaluated, following the removal of ferulenol, on three cell lines (MCF-7, HeLa and Saos-2). First of all, the results obtained have demonstrated that FER-E possesses an estrogenic-like property exerting a hyperproliferative effect, justified by the cellular expression of the receptor for estrogen [60,61]. Subsequently, in order to compare the effects of 17-β-E2 to those obtained with FER-E, we treated our cell lines with increasing concentrations of FER-E. The results surprisingly showed a dose-dependent FER-E effect: while lower concentrations (0.1–1.6 μM) demonstrated hyperproliferative effects, higher doses (3.2–50 μM) were toxic. The concentration of FER-E 0.8 μM, is the highest and the last that demonstrates non-toxic properties. This effect has also been previously demonstrated in neurons and oligodendrocytes [62], and is caused by the permeability of the biological membranes to cationic species, such as calcium and magnesium, which, in a dose-dependent way, undergo modifications on a part of the sesquiterpenes [48,63]. In fact, the toxicity of FER-E may also be due to its ability to mobilize the calcium ion and induce apoptosis by activation of caspase 3 [64,65,66]. These data are also confirmed by mortality results: in fact, from the concentration of FER-E 1.6 μM upwards, we appreciate cell mortality. The detected toxicity is not attributable to the presence of ferulenol, which is completely removed, but to the bioactive compounds found in the extract, which exert cytotoxic activity only at high concentrations. For example, the hexane and chloroform fractions, which are present in many extracts from Ferula genera, are responsible for cytotoxicity when their extracts are used at high concentrations [67,68]. Taken together, these results look very promising and suggest two important uses of FER-E: (a) *F. communis* can be used for “protective” purposes at low concentrations, when it is necessary to exert a hyperproliferative effect and in many disorders related to estrogen reduction. In menopause, for example, *F. communis* acts as a perfect phytoestrogen, capable of replacing HRT and avoiding its side effects. (b) *F. communis* can be used for “offensive” purposes at high concentrations, as an adjuvant along with chemotherapy in cancer treatment [69,70,71]. Further studies in clinical practice are needed, both in the short and long term, in order to increase knowledge about *F. communis* and its extract.

## 4. Conclusions

If new tests produce positive results, *F. communis* could become a good pharmacological aid to be used, in dependent concentration mode, as a phytoestrogen or for anticancer treatment. In addition, it would be necessary to monitor and regularize the growth of this plant that, to date, has a random expansion (it grows along the roadside and in uncultivated fields). Finally, extraction could be carried out with standardized protocols and knowledge of its bioavailability could be increased.

## 5. Materials and Methods

### 5.1. Obtaining and Characterizing of FER-E

FER-E was obtained from the root of the plant *F. communis**,* collected in Macomer, a small Italian town of the province of Nuoro, (Sardinia, Italy); this resort is located 563 m above sea level, at the foot of the Marghine chain. The root of *F. communis* is very deep in the soil and is covered by various secondary or adventitious roots, which branch laterally. The extraction protocol was as follows: 25 g of root were mixed with 125 g of acetone (1:5 ratio). The solution obtained was left at rest for 60 min to optimize the yield of the extraction process. After the time allowed, a filtration process of the solution was carried out to separate the liquid fraction from the solid portion rich in residues and impurities. The clarified liquid extract was dried to powder in a rotary evaporator under vacuum conditions, at a T of 40 °C. The amount of dry powder obtained was 2.2 g. Subsequently, a sample of filtrated solution was subjected to High Pressure Liquid Chromatography (HPLC). In particular, from the knowledge of retention time, the area and the height of the peak highlighted were extrapolated valuable qualitative and quantitative information. High-Pressure Liquid Chromatography (HPLC) analysis was performed on a Perkin Elmer Flexar Module equipped with a photodiode-array (PDA) detector, a series 200 autosampler, a series 200 peltier LC column oven, a series 200 LC pump, and an Agilent 4 μm C18 100A (250 × 4.6 mm) column. The control of the HPLC system and data collection was accomplished online by a computer equipped with Chromera software (version 3.4.0.5712). The reference standards were purchased by Sigma-Aldrich. An amount of 10 mg of Ferula roots extract dry powder was dissolved in 10.0 mL of methanol. The resulting solution was vortexed for complete dissolution. The solution was filtered with a 0.2 μm PTFE filter and 10 μL of sample were injected into the HPLC system. A two-solvents gradient (0.88% trifluoroacetic acid/acetonitrile) was used for the elution with a flow of 1 mL/min keeping the column at 30 °C. The wavelength of the detector has been set to 256 nm. A further chromatographic analysis has allowed to estimate titre pairs to 25% in ferutinin.

### 5.2. Cell Cultures and Treatments

The human breast cancer cell line (MCF-7, CVCL 0031; range of passages = 66 to 87; passage number 73), human cervical cancer cell line (HeLa, CVCL 0030; range of passages = 70 to 164; passage number 90) and human osteosarcoma cell line (Saos-2, CVCL 9U47; range of passages = 10 to 85; passage number 25) were purchased from the American Type Culture Collection (20099 Sesto San Giovanni, Milan, Italy). These cell lines were cultured in Dulbecco’s modified Eagle’s medium (DMEM) supplemented with 10% heat-inactivated fetal bovine serum (FBS), 100 U/mL penicillin and 100 µg/mL streptomycin in a humidified 5% CO_2_ atmosphere at 37 °C. When cells reached a 50% confluence, they were treated with FER-E, or 17-β-E2 (24 h, 48 h and 72 h) and then the appropriate tests were carried out. 

### 5.3. Proliferation Assay and Cytotoxicity Study

Cell proliferation was assessed by colorimetric assay 3-(4,5-dimethylthiazol-2-yl)-2,5-diphenyltetrazolium bromide (MTT) and this test is based on the observation that the mitochondria of viable cells possess active enzymes capable of reducing MTT to a dark blue visible reaction product form. Therefore, a reduction of MTT, measured colorimetrically, provides information on the cellular metabolic activity and the viability. Cells were seeded into 96-well plates (8 × 10^3^ cells/well) and on the next day, the growth medium was replaced with a medium containing FER-E or 17-β-E2 as described above. Then, the medium was replaced with phenol red-free medium containing MTT solution (0.5 mg/mL) and, after 4 h incubation, 100 μL 10% SDS was added to each well to solubilize the formazan crystals. The optical density was measured at wavelengths of 540 and 690 nm by means of a spectrophotometer (X MARK Spectrophotometer Microplate Bio-Rad). 

The data obtained with the cell proliferation assay have been enriched and confirmed by the measurement of cytotoxicity by cell exclusion assay of trypan blue (0.4% *w*/*v*). This dye can only penetrate cells if the plasma membrane is damaged. As a result, dead cells will appear in blue and be distinguishable from living cells by microscopic analysis. Cell death was reported as the percentage of stained (non-viable) vs. total cells counted [20]. In the experiment, cells were seeded in 12-well plates at a density of 50 × 10^3^.

### 5.4. Annexin V Staining

The cells, after being treated as described, were detached using trypsin, washed twice with cold PBS and suspended in binding buffer (Metabolic activity/Annexinv/Kit for dead cell apoptosis) at a concentration of 1 × 10^6^ cells/mL. Then, 100 µL of the suspension was transferred to 5 mL tube and 5 µL of FITC Annexin V (BD Biosciences, San Jose, CA, USA) was added. The samples were vortexed gently and incubated for 15 min at 25 °C in the dark. Finally, 400 µL of 1x binding buffer and 5 µL of propidium iodide (PI) were added to each tube and incubated for 1 h; the samples were then analyzed by flow cytometry (emission filter 515–545 nm for FITC; 600 nm for PI). In total, 20,000 cells per sample were acquired using the FACS Accuri (Becton Dickinson, Milan, Italy) laser flow cytometry.

### 5.5. Statistical Analysis

The results are expressed as means ± standard deviation (SD) and evaluated statistically for difference by one-way analysis of variance (ANOVA), followed by Tukey–Kramer test for multiple comparisons. A value of *p* < 0.05 was considered to be significant. Statistical analysis was conducted using GraphPad Prism version 5.0 statistical software (GraphPad Software Inc., San Diego, CA, USA).

## Figures and Tables

**Figure 1 plants-11-01905-f001:**
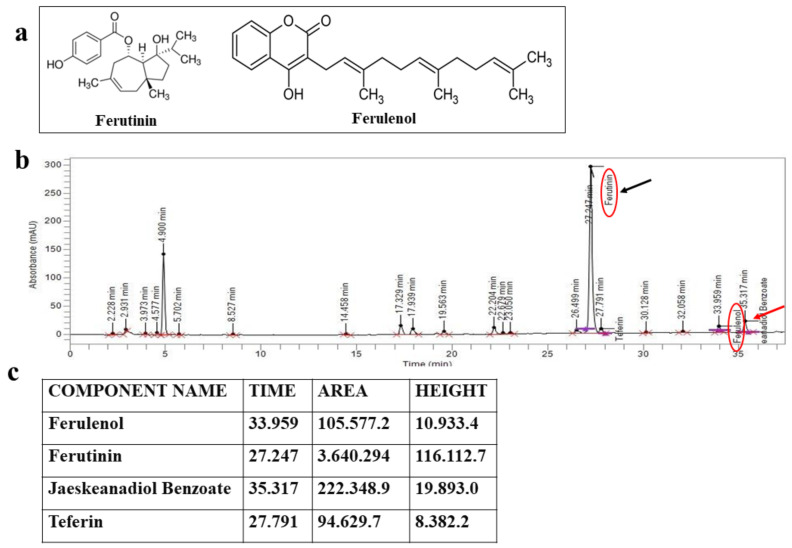
In panel (**a**) the chemical structures of ferutinin and ferulenol are represented; in panel (**b**) HPLC spectrum of FER-E, in which ferutinin and ferulenol are present, is shown. In panel (**c**) HPLC analysis values are represented.

**Figure 2 plants-11-01905-f002:**
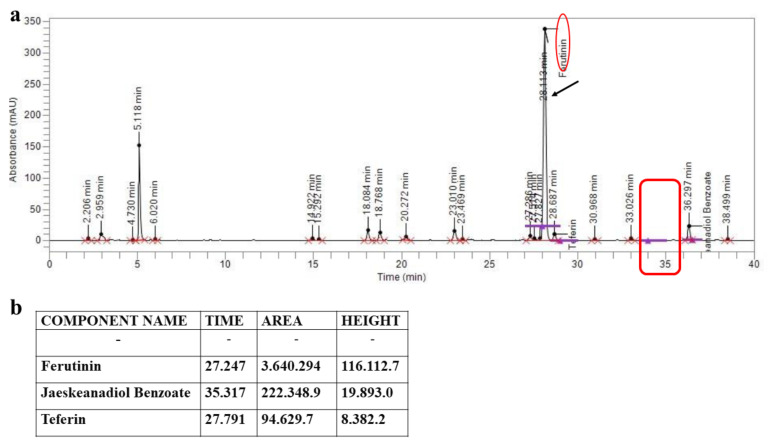
HPLC spectrum of FER-E in which ferulenol is absent. In panel (**a**) the HPLC spectrum of FER-E is shown. In panel (**b**), HPLC analysis values are represented.

**Figure 3 plants-11-01905-f003:**
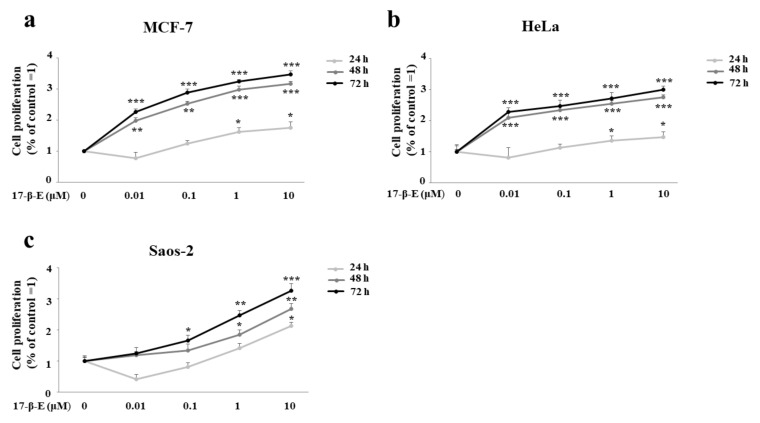
Hyperproliferation induced by treatment with 17-β-E2 in MCF-7, HeLa and Saos-2 cell lines. Cell lines were treated with increasing concentrations of 17-β-E2 (0.01–10 μM) and the proliferation rate was measured through the MTT test. Each panel is composed of three curves obtained from the data measured at 24, 48 and 72 h of treatment. In particular, panel (**a**) shows the results related to the cell line MCF-7; panel (**b**) refers to HeLa cells; and finally, panel (**c**) highlights the data related to Saos-2 cells. Three independent experiments were carried out, and the values are expressed as the mean ± standard deviation (sd). * denotes *p* < 0.05 versus the control; ** denotes *p* < 0.01 versus the control; *** denotes *p* < 0.001 versus the control. Analysis of Variance (ANOVA) was followed by a Tukey–Kramer comparison test. Each treatment was carried out in quadruplicate.

**Figure 4 plants-11-01905-f004:**
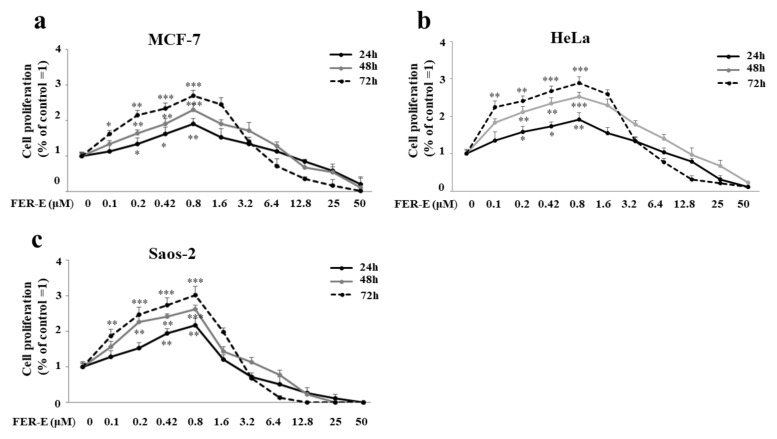
Dose-dependent effect induced by FER-E treatment. Dose–response curves were constructed by treating the cell lines considered with increasing concentrations of FER-E. Panels (**a**–**c**) refer to viability data induced in MCF-7, HeLa e Saos-2, respectively. Three independent experiments were carried out and the values are expressed as the mean ± standard deviation (sd). * denotes *p* < 0.05 versus the control; ** denotes *p* < 0.01 versus the control; *** denotes *p* < 0.001 versus the control. Analysis of Variance (ANOVA) was followed by a Tukey–Kramer comparison test. Each treatment was carried out in eightfold.

**Figure 5 plants-11-01905-f005:**
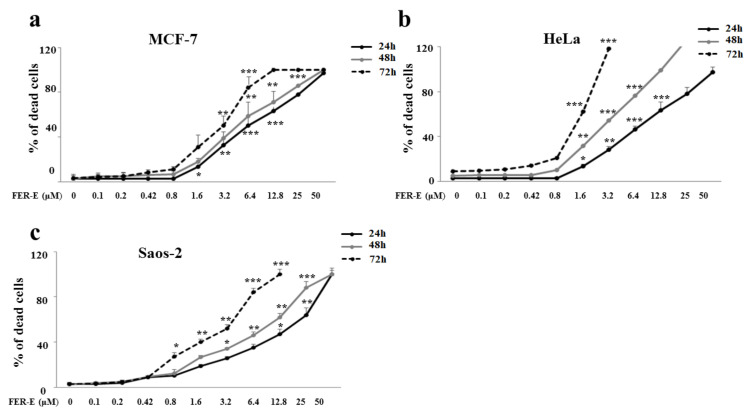
Percentage of mortality induced by treatment with FER-E. Percentage of mortality was calculated following treatment with FER-E for 24, 48 and 72 h by the Trypan blue test. In particular, mortality induced by treatment with FER-E on MCF-7, HeLa and Saos-2 cell lines was shown in panels (**a**–**c**), respectively. Three independent experiments were carried out, and the values are expressed as the mean ± standard deviation (sd). * denotes *p* < 0.05 versus the control; ** denotes *p* < 0.01 versus the control; *** denotes *p* < 0.001 versus the control. Analysis of Variance (ANOVA) was followed by a Tukey–Kramer comparison test. Each treatment was carried out in triplicate.

**Figure 6 plants-11-01905-f006:**
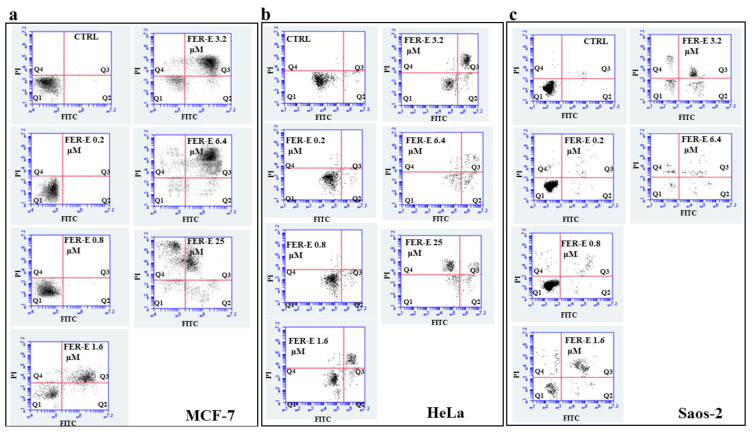
Mortality assessed by Annexin V Staining assay. Figure 6 shows the effect of treatment with different concentrations of FER-E on cell lines. In particular, in panels (**a**–**c**) the results for lines MCF-7, HeLa and Saos-2 are represented, respectively. Cytometric analysis was conducted on 20,000 events and each treatment is represented by a dot plot divided into 4 quadrants (Q1, Q2, Q3, and Q4). Q1 refers to Annexin V-negative/PI-negative cells (viable cells). Q2 refers to Annexin V-positive/PI-negative cells (apoptotic cells). Q3 refers to Annexin V-positive/PI-positive cells (late apoptosis). Q4 refers to Annexin V-negative/PI-positive cells (advanced necrosis). A representative experiment of three independent experiments was shown.

**Figure 7 plants-11-01905-f007:**
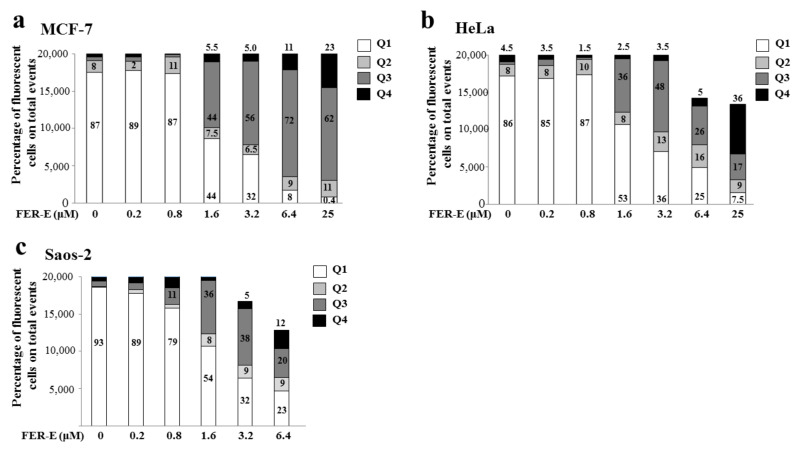
Expression of results obtained by Annexin V Staining assay. The results of three independent experiments, obtained by Annexin V Staining assay are shown. In particular, the percentage of cellular fluorescence in the total number of events (20,000) has been evaluated and indicated in the relevant box. In panels (**a**–**c**) the quantifications of MCF7, HeLa and Saos-2 cell lines are shown respectively. Data from an indicative experiment (of 3 independent experiments with similar results) are shown.

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
