# Peer review of "Ferula communis L. (Apiaceae) Root Acetone-Water Extract: Phytochemical Analysis, Cytotoxicity and In Vitro Evaluation of Estrogenic Properties"

_plants, 2022, doi:10.3390/plants11151905_

Round 1
Reviewer 1 Report
Manuscript entiteled “Ferula communis L. (Apiaceae) root acetone-water extract: phytochemical analysis, cytotoxicity and in vitro evaluation of estrogenic properties” review. I suggest a major revision before proceeding further.
Please find the following comments:
1. Note on the semantic discussion. I understand the Authors' intention to place the results in the context of menopause, but sentences like: "These cell lines are responsive to the action of estrogens and are involved during menopause" or "Subsequently, the effects of FER-E will be evaluated on three cell lines belonging to districts directly involved during menopause..."are extremely inaccurate. Cell lines only mimic certain processes that can occur in vivo, and we often extrapolate data obtained using cell lines to in vivo processes. However, explicitly emphasizing the involvement of cell lines in the menopausal process is misleading and introduces confusion to the reader. The example sentences listed above should be avoided. Please note this context throughout the text and then correct it as appropriate.
2. Are the pictures shown in Figure 1 the property of the Authors from personal research? If so, add appropriate information to the legend of this figure, and if not, provide the source of this data.
3. In M&M part please provide original numbers characterizing the cell lines used, the range of passages from which the cells were used for the research.
4. In M&M part, the statement "live cells possess mitochondria" is unfortunate - please correct it.
5. Please briefly describe the test procedure with trypan blue.
6. On Fig.6 please use “% of dead cells”, “Cytotoxicity”, or else instead of “% cell death”
7. What was the proliferation and viability of cells at 17-b-E concentrations higher than 10 µM? If it was tested and not added to the results, please complete or add a relevant comment.
8. The subsection titles in the results are too general. Please detail.
9. The section on statistical analysis is missing from the M&M section. It should be completed how exactly the analysis was performed, what tests were used, what software was used.
10. The repeat count information for viability and proliferation tests was omitted. Only the number of three independent experiments is indicated. Please enter the number "n" of repetitions that were used for the statistical calculations.
11. In the graphs shown in Figures 4 and 5, please convert and change the signatures of the Y axis to those more relevant to the data, i.e. cell proliferation (% of control).
12. In Figure 7 panel c, no data for the 25uM concentration. Please add to make the figure consistent. This is surprising, especially since in the next figure there are quantitative data corresponding to results not shown.
13. I suggest that we remove Figure 8 from the text - it duplicates the results in Figure 7. Instead of Figure 8, I suggest that the Authors add the percentage of cells (not the percent of fluorescence) in the individual squares to Figure 7. Please add the percentages to the respective gates while removing the description of the condition in the graphs so as not to disturb the readability. The description of the condition can be placed above the graph.
14. The graphs in Figure 7 should be larger and clearer. Maybe consider placing them for each line from top to bottom. Please add the appropriate staining on the x and y axes (instead of the fluorescence channel). Delete any unnecessary work markings. The quality of this figure in this shape is poor.
15. The resolution of Figures 2b and 3 is too low, making the data unreadable.
Author Response
Dear reviewer,
I send in attachment the answers to your suggestions. Only the correction of English is missing because I sent the request to the English editing of Plants, but my department is late with the payment. As soon as these bureaucratic problems are resolved, I will send you the manuscript again with the certification of the control of English.

Reviewer 2 Report
Dear authors,
The subject of the manuscript is interesting, but the text needs improvement. I suggest а minor revision of the manuscript. My remarks do not concern the methodology and the results obtained, but the writing of the article. The improvement of the English language is needed, concerning the grammar, stylistics and technical mistakes, as well. In addition, the authors have not complied with the journal’s requirement to structure the manuscript. Materials and methods should be placed after Discussion. The article should end with a Conclusion.
In Introduction
The introduction needs to be rewritten in order to have well-structured background information for the proposed research. The introduction should contain data on whether the root extract from the study plant (Ferula communis L.) has been tested on examined cell lines (MCF-7 cells; HeLa cells; Saos-2 cell). Instead there is unnecessary information such as botanical description of the species.
Page 2 line 66 Several data highlighted that Ferula communis L. extract (FER-E), completely deprived of ferulenol, could be an effective alternative hormone therapy to prevent and treat post-menopausal symptoms thank to high amount of the sesquiterpene ferutinin. – The sentence should end with citations.
It is not necessary to write the full botanical name Ferula communis L of the plant species throughout the text, only at the first mention. It should be F. communis in the following mentions.
In Materials and Methods
Please write what the abbreviation17-β-E2 means. (17-β-estradiol?)
In Discussion
There are many repetitions – for example that ferunelol can cause ferulosis – It is written in the Abstract, in Introduction and in Discussion. The Discussion needs to be enriched with other research of the topic.
The Conclusion should be in a separate section.
Author Response
Dear reviewer,
I send in attachment the answers to your suggestions. Only the correction of English is missing because I sent the request to the English editing of Plants, but my department is late with the payment. As soon as these bureaucratic problems are resolved, I will send you the manuscript again with the certification of the control of English

Reviewer 3 Report
There are many errors and major issues in the manuscript.
Abstract:
Authors need to present findings as integral values or percentage.
Line 20, 25, 26, 28, 34, 36,
but also from leaves and rhizome…. Remove “from”
This compound is capable of perform.. check the sentence
cointaining high amount of ferulenol.. check the spelling
, determins a double… check
in addition to possess … check
(1,6-50 μM) were toxic…check – 1.6? and revise in other part of the manuscript too. This problems
Can be seen in other section as well.
Introduction
---Information is insufficient on peer-findings for highlighting the research and knowledge gaps. Authors have not established study rationale/justification through critical analyses of peer-findings. Write the importance of phytochemicals such as phenolic compounds and their estrogenic properties.
Line 44, 45, 47, 48
present in all of Central Asia… add “of”,
ancient latin name…. make it Latin
1-2,5 m high,… check
stem which that elongates
Label the each figure. Do not submit the picture, if it is downloaded from Google or from other published paper.
---The hypothesis is missing.
Material and methods
Line 80 and 82, 97, 130
a little italian town.. Check
is a strong and very
The detector wave length…. Check
) were added for 1 h
Results
Check your data
Line 160—166, 183—186, 218
Discussion
Adjust the line-247-261 in the introductory section. Avoid repetition.
Authors need to strengthen the discussion section by adding more interpretations of recorded findings supported by peer-findings.
Include the mechanism in the discussion.
Do not mention the figures and tables number in the discussion section as they were already mentioned in the result section.
Use recent and related literature to discuss your findings.
References:
Some references are not in journal format.
Author Response
Dear Reviewer,
I send in attachment the answers to your suggestions. Only the correction of English is missing because I sent the request to the English editing of Plants, but my department is late with the payment. As soon as these bureaucratic problems are resolved, I will send you the manuscript again with the certification of the control of English

Round 2
Reviewer 1 Report
All comments were taken into account. I have no more comments.
Reviewer 3 Report
The author of the manuscript has revised the research article satisfactorily and responded to all the comments. Now the manuscript can be accepted.